# DIALOR (DIgitAL cOaching for fRailty): protocol for a single-arm mixed-methods feasibility study of a digital health coaching intervention for older people with frailty in primary care

Matthew I Sait ,[1] Rachel A Christie,[1] Chantel Cox,[1] Michele Board,[1] Sarah Thomas ,[1] Cheryl O'Sullivan,[2] Cheryl Davies,[3] Dawn-Marie Walker,[4] Michael Vassallo,[5] E A Sadler,[6] Mark Allen-Pick,[1] Patrick Moore,[7] Katherine Bradbury,[8] Jane Murphy[1]

For numbered affiliations see end of article.

**Correspondence to**
Matthew I Sait;
saitm@bournemouth.ac.uk

## ABSTRACT

**Background** Multidomain interventions in older adults offer the best opportunity to prevent, delay or reverse existing symptoms in the earlier stages of frailty and improve independence but can be costly, and difficult to deliver at scale. However, digital health interventions enable personalised care and empowerment through self-management of long-term conditions, used at any time and when combined with health coaching offer the potential to enhance well-being and facilitate the achievement of health-related goals. We aim to evaluate the feasibility and acceptability of a digital health platform for long-term disease management combined with health coaching for people living with mild-moderate frailty, targeting self-identified goals—activity, nutrition, mood, enhancing social engagement and well-being.

**Methods and analysis** This is a non-randomised feasibility, single-group, pretest/post-test study, using qualitative and quantitative methods. The digital health coaching intervention (DIALOR—DIgitAL cOaching for fRailty) has been developed for implementation to older adults, aged 65 years or older with mild to moderate frailty and diagnosis of one or more long-term health conditions in the community. Participants will receive 12 weeks of health coaching and have access to a mobile health platform for 6 months. The primary outcome measure is the acceptability and feasibility of DIALOR along with a range of secondary outcome measures (including frailty, functioning measures, quality of life, social engagement, diet quality and self-reported indicators) collected at baseline and at 6 months. The findings will inform whether a wider effectiveness trial is feasible and if so, how it should be designed.

**Ethics and dissemination** Ethical approval has been granted by the Southeast Scotland Research Ethics Committee 02 (reference: 22/SS/0064). Research findings will be disseminated in a range of different ways to engage different audiences, including publishing in open-access peer-reviewed journals, conference presentations, social media, dissemination workshop with patients, carers, and healthcare professionals and on institution websites.

## STRENGTHS AND LIMITATIONS OF THIS STUDY

⇒ A strength of the study design is using a mixed-methods approach with a qualitative understanding of the implementation process and quantitative measures to assess the effects on health outcomes and whether the trial is feasible.
⇒ There is strong involvement of stakeholders and people with the lived experience to develop and guide the study.
⇒ Potential limitations are as a feasibility study, it will assess a pretest and post-test intervention with non-randomisation of sample and to sufficiently engage hard-to-reach target groups including those with poor digital literacy and low socioeconomic status, impacting representation of the participant group and generalisability of the findings.

## INTRODUCTION

Frailty is a multidimensional syndrome characterised by cumulative loss of reserves and accumulation of deficits (eg, energy, physical ability, cognition, social and psychological aspects) that give rise to vulnerability and risk of adverse outcomes with advancing age.[1] It is not an inevitable consequence of ageing but older adults with frailty have an increased risk of poor health, including falls, fractures and functional impairment, hospitalisation, need for nursing home care, and increased mortality.[2] Older people with frailty experience impairment in their activities of daily living and often report a significant reduction in their quality of life.[3] Furthermore, studies have shown that frailty may be related to decreased mood[4] and may be shaped by social factors such as different levels of social support or living alone.[5] Frailty affects about 12% of people aged ≥65 years worldwide[6] and

14% of older people in the UK,[7] and even higher numbers of individuals in the early stages of frailty, ranging from 19% to 53%.[8] Frailty is an ever-present burden among international populations, which has been exacerbated by the impact of the COVID-19 pandemic[9] and the resultant deconditioning.[10]

While moderate to severe frailty has a higher risk of physical health declines, mild to moderate frailty with some loss of physiological reserve is considered to be potentially reversible[11] to a robust or stable state.[12] Mild frailty is conceptualised as an intermediate stage on the Rockwood Clinical Frailty Scale,[13] whereby older people experience some loss of physiological reserves but can usually recover after a negative stressor event, such as an infection or fall. This would indicate that interventions targeting earlier stages of frailty are more likely to reverse frailty decline. Moreover, there is evidence to suggest that interventions, particularly multidomain interventions in older adults including mobility, strength, balance, nutrition, physical activity and social support offer the best opportunity to prevent, delay or reverse existing symptoms of frailty and improve independence.[14–17] Most multicomponent interventions incorporate involvement from a multidisciplinary team of health and social care professionals meaning they can be costly and difficult to deliver at scale.[18]

Digital health interventions (DHIs) enable personalised care and empowerment through self-management of a long-term condition (LTC), which can be used at any time, and may offer more cost-effective healthcare delivery.[19] DHIs have been shown to be effective at achieving successful remote monitoring and management of chronic obstructive pulmonary disease (COPD) among older populations,[20] suggesting that DHIs may be a feasible method of reaching remote communities. Studies show that digital approaches for lifestyle modifications in other LTCs such as type 2 diabetes, offer effective and scalable options when face-to-face or in-person programmes are not accessible or feasible, especially when offered with health coaching (HC).[21 22]

HC was originally described as the practice of health education and health promotion within a coaching context, to enhance the well-being of individuals and to facilitate the achievement of health-related goals.[23] More recently, HC has been defined as helping people gain and use the knowledge, skills and confidence to become active participants in their care so that they can reach their self-identified health and well-being goals.[24] In the last decade, HC has become increasingly embedded within integrated care services. However, the evidence base demonstrating the effectiveness of HC in the UK is limited. For example, a systematic review and meta-analysis assessing the impact of HC on type 2 diabetes found that HC was effective in achieving optimal glycaemic levels via self-management strategies.[25] In the UK, health coaches working within primary care can be established clinical team members (eg, registered nurses, physician associates, physiotherapists, paramedics, occupational therapists and psychologists) or can be unlicensed health workers.[26] Furthermore, research is emerging to show that HC may offer a promising approach to support effective lifestyle interventions for managing chronic conditions such as asthma, diabetes, cardiovascular disease (CVD) and mental illness on an international stage.[27]

Current integrated care systems in the UK have also implemented both DHIs via mHealth and HC to improve self-management of their LTCs, among older people living with COPD, diabetes and CVD.[28] However, there remains a distinct lack of evidence, which demonstrates the effectiveness of HC for older adults who have at least two chronic conditions (multimorbidity), which is thought to affect up to 50% of older people (aged over 65 years).[29 30] Further, the impact of HC for older adults with a diagnosis of frailty, is unknown, making a single disease-focused model of healthcare unsuitable to provide effective management.[31] Therefore, targeted behaviour change interventions using DHIs such as mHealth, in conjunction with HC, have the potential to offer a proactive and wide-reaching approach to prevent frailty decline alongside the management of other diagnosed comorbidities and improve health and well-being outcomes.

This study aims to address this gap by codesigning and implementing a multicomponent intervention for people living with mild to moderate frailty. The DIgitAL cOaching for fRailty (DIALOR) intervention will include two key components (figure 1):

► Digital HC
  – This will include digital HC sessions with a qualified health coach. An online platform (Zoom/Accurx/Microsoft Teams) will be used to deliver the digital HC intervention.
► Mobile health platform
  – A suite of web-based platforms offering tailored clinical advice and support and self-management resources for the management of COPD, asthma, CVD and type 2 diabetes (details of the online platform can be accessed via https://mymhealth.com/). The platforms can be accessed by any electronic device (eg, smartphone, tablet or television) that has an active internet connection.

The primary objectives of this study are as follows:

► Explore barriers and facilitators of the digital intervention from the perspectives of health coaches and healthcare professionals (HCPs) and determine the essential and desirable features of the intervention (wants and needs; preferred outcomes) to identify what functions will be most useful for the target population.
► Explore how HC can provide support for the management of mild to moderate frailty and LTCs from the perspectives of older people with mild to moderate frailty, carers, HCPs and health coaches.
► Determine the acceptability and feasibility of the intervention for older adults with mild/moderate frailty and another LTC referred to access HC services and willing to engage with the mobile health platform.

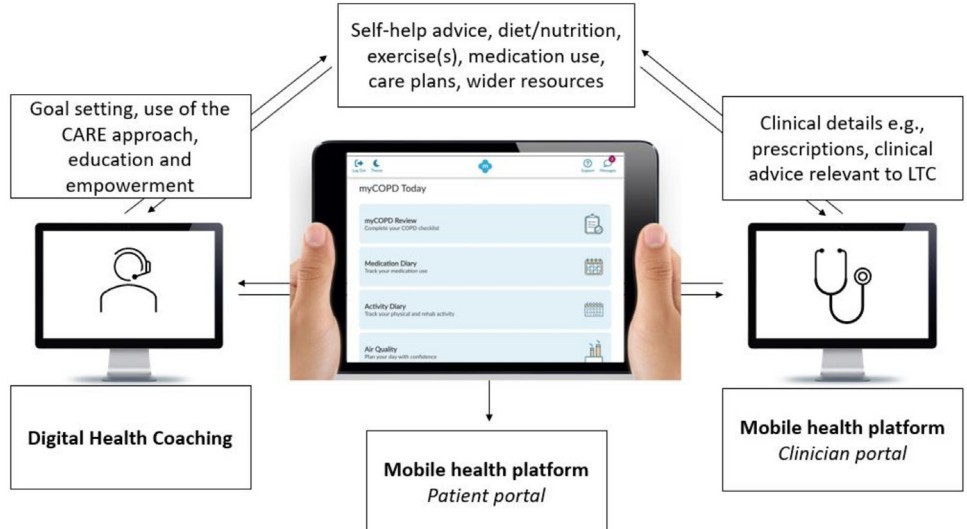

**Figure 1** Overview of the DIALOR intervention and its functionalities. CARE, Congratulate, Ask, Reassure, Encourage; COPD, chronic obstructive pulmonary disease; DIALOR, DIgitAL cOaching for fRailty; LTC, long-term condition.

The secondary objectives of this study are as follows:
► Explore the factors which inhibit or facilitate implementation including contextual factors, recruitment processes, resources needed, potential mechanisms and pathways to impact.
► Explore the acceptability and suitability of outcome measures (to inform sample size estimation and determine the primary outcome measure) for a future randomised controlled trial of the intervention (should it be feasible).
► Measure key domains for secondary outcome measures including participants who are able to self-report measures.

## METHODS AND ANALYSIS
### Intervention development
Over a 9-month period, intervention development will use guiding principles[32] to inform codesign objectives of a logic model (figure 2) and ensure the intervention is acceptable, feasible, engaging and easy to implement. A systematic review and narrative synthesis will also guide the development of key features our intervention requires to meet user needs and best practice guidelines.

Behavioural analysis[33] will triangulate evidence from (1) the narrative synthesis, (2) consultation with two stakeholder engagement groups (SEG) including HCPs, health coaches, commissioners of service and patients and (3) patient and public involvement (PPI) advisors and (4) a separate PPI group with lay advisors to identify barriers in each domain—activity/mobility, nutrition, social engagement and psychological well-being. Intervention components addressing each barrier will be theoretically mapped and characterised using the behaviour change wheel,[34] the theoretical domains framework[35] and normalisation process theory (NPT). NPT is the process through which complex, new ways of working become embedded into standard practice.[36] We will use the Behaviour Change Techniques Taxonomy Version 1 (BCTT V.1)[37] to code the behaviour change techniques delivering

the identified intervention functions. This will provide a clear description of the intervention and factors which may influence its implementation and will be summarised in a behavioural analysis table. We will then convene another SEG meeting to discuss and refine the proposed logic model.

To identify challenges/enabling factors for delivery/implementation and intervention modifications needed to maximise engagement and behaviour change, we will conduct individual interviews with people with mild to moderate frailty (and their carers where appropriate) (n=20) and focus groups (n=4) with HCPs and health coaches (n=20) (figure 3). We will undertake purposive sampling to select older participants (following ethics protocol for consent), up to 20 semistructured interviews or until data saturation is reached, in accordance with robust methods in qualitative research.[38] These will be conducted face to face, and audio recorded using dictaphones, otherwise, they will be conducted and recorded through Microsoft Teams or Zoom. Final in-house testing of the intervention will be undertaken before the commencement of the study. We will use the the Template for Intervention Description and Replication (TIDieR) checklist[39] for intervention description which will report relevant outcomes to inform further planning for a future trial.

### Study design
This feasibility study is a non-randomised, single-group, pretest/post-test study, using qualitative and quantitative methods. This protocol was developed in accordance with the Standard Protocol Items: Recommendations for Intervention Trials reporting guidelines.[40]

This will be a mixed-methods prospective cohort study conducted within primary care settings in Southwest England. First, online group training on the intervention will be provided for health coaches by the research team, with top-up training and support as necessary throughout. A training manual will also incorporate frailty awareness and the Congratulate, Ask, Reassure, Encourage model

| Problem | Intervention targets | Intervention ingredients | Mechanisms to be examined in process analysis | | Outcomes |
|---|---|---|---|---|---|

**Figure 2 logic model content:**

**Problem:** Increased frailty

**Intervention targets:**
- Physical activity
- Diet
- Independent in completing activities of daily living (ADLs)
- Social engagement
- Support confidence in change

**Intervention ingredients:**

Health coaching:
- Motivation interviewing
- SMART goals
- Increased physical activity
- Nutrition guidance
- Challenge illness perceptions
- Guidance for using digital platform

Digital platform:
- Specific education and guidance for long-term conditions
- Exercise videos
- Nutrition guidance
- Place to record progress

**Mechanisms to be examined in process analysis:**

Participant
- Engagement with health coach
- Perceptions of engagement with health coach
- Engagement with digital platform
- Perceptions of engagement with digital platform
- Satisfaction

Health coach, health care professionals, commissioners
- Engagement with intervention
- Perception of intervention
- Barriers and facilitators to intervention

Behaviour change:
- Increased physical activity
- Improved diet
- Confidence to do things independently
- Adherence to change
- Reasons for non-adherence

**Outcomes:**

Primary outcome: Acceptability and feasibility

Secondary outcomes:
- Function
  - Modified Barthel Index
- Physical activity
  - IPAQ
- Wellbeing
  - Warwick Edinburgh Mental Wellbeing Scale
- Quality of life
  - EQ-5D-5L
- Independence in ADLs
- Frailty severity
  - Groningen
- Self-reported measures e.g., social engagement measure, digital literacy, reported diet/activity goals, number of falls, alcohol intake, smoking status
- Weight loss
- BMI
- Short form FFQ

**Figure 2** Proposed logic model. BMI, body mass index; EQ-5D-5L, EuroQol-5 dimensions-5 levels; FFQ, Food Frequency Questionnaire; IPAQ, International Physical Activity Questionnaire.

approach.[41 42] Intervention delivery will be tailored to each patient, with goals and strategies to help achieve them developed in conjunction with the health coach.

### Study setting

The study will work with GP practices in Primary Care Networks in Southwest England to identify patients already using the mobile health platform. GP practices will be eligible to take part if they are currently referring older people with mild to moderate frailty to access HC/social prescribing services and are willing to test new digital approaches.

### Participant recruitment

Participant recruitment will be conducted within primary care services, at individual participating GP surgeries. Up

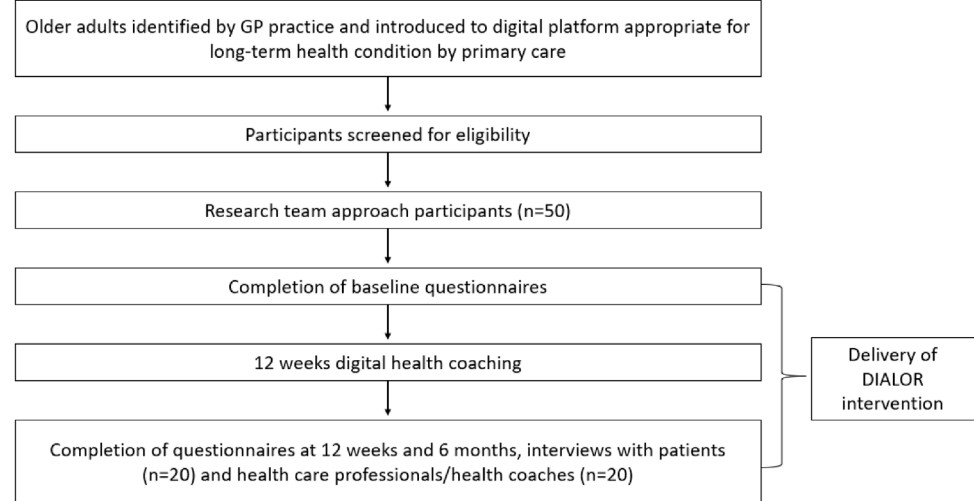

**Figure 3** Process flow of study design. DIALOR, DIgitAL cOaching for fRailty; GP, general practitioner.

## Box 1  Inclusion and exclusion criteria

### Inclusion criteria
⇒ Aged ≥65 years.
⇒ Scoring as mild or moderately frail using the Electronic Frailty Index.
⇒ Diagnosis of one or more long-term conditions including chronic obstructive pulmonary disease, type 2 diabetes, heart disease or asthma.
⇒ Living in their own homes.
⇒ Capacity to consent to participate.
⇒ Able to communicate fluently enough in English to take part in the study.
⇒ Able to recall their experiences sufficiently to engage with a health coach.
⇒ Self-reported ability to use digital technology independently or with support.

### Exclusion criteria
⇒ Nursing home resident or on the active waiting list for a place.
⇒ Individuals who have a terminal illness and are being managed for end-of-life needs.
⇒ Lacking the capacity to consent (eg, advanced dementia).

to 50 older people with mild to moderate frailty will be recruited in line with previous sample size recommendations for feasibility studies.[43] Eligibility criteria for older adults with mild to moderate frailty to participate in the study are outlined in box 1. The clinical care team from participating GP surgeries will identify eligible participants through existing medical records.

Three strategies will be used to recruit patients (1) via text message invitation to eligible patients which provides a link to a website with further information about the study, participant information sheet and contact information for the research team. Potential participants will be followed up by the research team to provide an opportunity to answer questions, confirm eligibility and obtain informed consent to participate, (2) by a member of the research team approaching eligible participants at the GP surgery, signposted by the clinical team and (3) via leaflets provided to eligible patients by the clinical team.

### Digital support for patients
Patients with low digital literacy will be supported by the research team to access the digital HC and mobile health platforms. They will be offered a training manual, developed by the research team and coproduced alongside the SEG and PPI group alongside digital support leaflets and guidance on keeping safe online. To reduce the risk of digital inequity, internet-enabled tablets may be provided to individuals who lack a device, for the length of the study. Internet and data packages (via local council schemes) will be provided if the participant lacks access to an established internet connection. Digital support will be offered throughout the study via local council services (volunteer digital upskilling clinics embedded within public spaces, eg, GP surgeries and council libraries) and the research team.

### Outcome measures
#### Primary outcomes
The primary outcome measures are the feasibility and acceptability of the DIALOR intervention. Feasibility will be determined from the number of health coaches trained, the number of participants recruited, retention and adherence to the intervention as well as any adverse events. To determine the acceptability of the intervention and to explore barriers and enablers to the implementation of the intervention, semistructured interviews will be conducted with patients with mild to moderate frailty (and their carers where appropriate) (n=20), HCPs and health coaches (n=20) who participated in the study.

The interviews will consist of open-ended questions to explore participant views of the intervention, barriers and facilitators to inform the design of further studies. To understand the factors influencing implementation, the interview schedule will be underpinned by NPT.[44] NPT is an implementation theory providing a framework to identify and explain important elements of the implementation process. As such, it informs an understanding of the social processes through which new or practice can be implemented, embedded and integrated into care settings.

#### Secondary outcomes
The secondary outcomes will include the measurement of functioning, quality of life, physical activity levels, frailty severity, diet quality and engagement using validated outcome measures.

Physical functioning, a key outcome in frailty trials,[45] will be measured using the Modified Barthel Index (MBI)[46] and Lawton-Brody Instrumental Activities of Daily Living (IADL).[47] The MBI will measure older adults' functional ability using 10 items, with a higher score reflecting the higher ability to function independently. The MBI has been found to be of reasonable reliability when used among older adults.[48] The IADL will measure more complex activities of daily living using eight key domains. The International Physical Activity Questionnaire short form (IPAQ-SF) will measure older adults' activity levels in the last 7 days.[49] The evidence supporting the IPAQ-SF as an indicator of relative or absolute physical activity is weak.[50] However, it is a validated generalised outcome measure for older adult populations (65 years or over).[51]

Quality of life is a core outcome in frailty trials[52] and will be measured using the EuroQol-5 Dimensions-5 Levels (EQ-5D-5L) questionnaire, which includes five health areas (mobility, self-care, usual activities, pain/discomfort and anxiety/depression). These five health areas are assessed with five main response levels (no problems, slight, moderate, severe or unable), along with a Visual Analogue Scale.[53]

Frailty severity will be measured using the Groningen Frailty Indicator (GFI).[54] The GFI is a multidimensional screening instrument consisting of 15 self-report items. It is a feasible and widely used screening instrument for identifying frail older adults.[55 56] Higher GFI scores

indicate higher frailty levels and an increased need for integrated care.[57]

The Short Form Food Frequency Questionnaire (SFFFQ) will measure dietary quality. The SFFQ asks about foods which respondents might have consumed, on average, over a set time period (3 months) and is found to be an effective instrument for assessing diet quality in large-scale studies with UK adult populations.[58]

Social engagement will be measured using the Lubben Social Network Scale (LSNS-6) and mental well-being will be measured using the Warwick-Edinburgh Mental Well-Being Scale (WEMWBS).

Self-reported indicators such as height, weight, body mass index, number of falls (per year), goal attainment, alcohol intake and smoking habits (if applicable) will be measured at baseline (preintervention), throughout the study during digital HC intervention and at 6 months (postintervention).

Data will be collected from the mobile health platform which will be recorded automatically by inbuilt tracking software, including usage patterns from the number of logins, and the length of time that older people and their carers use the platform. These data will be collected at 6 months for each patient participant (postintervention).

### Data analysis
#### Quantitative data analysis
Data will be entered into a secure database for analyses. Statistical analyses will be conducted using statistical software SPSS (V.28.0). Descriptive statistics: median (IQR); mean (SD); number (%) and CI—will be used to analyse the number of volunteers recruited and retained, as well as participants' adherence to the intervention and pre–post intervention outcome measures to assess the feasibility of delivering the intervention. Preanalyses and postanalyses at 6 months will be conducted to determine if the intervention had an impact on the lifestyle management of frailty measured by functioning (MBI, IADL), social engagement (LSNS-6), mental well-being (WEMWBS) quality of life (EQ-5D-5L), physical activity (IPAQ-SF), frailty indicator (GFI), diet quality (SFFFQ) and self-reported indicators.

#### Qualitative data analysis
Data collected from all the interviews and focus groups will be transcribed verbatim and analysed using thematic analysis (TA). TA is a commonly used method for identifying, analysing and reporting patterns or themes within data.[59] There are six key phases: phase 1—familiarisation with the data, phase 2—initial code generation, phase 3—searching for and generating themes, phase 4—reviewing themes, phase 5—defining and naming themes and phase 6—producing the report. Analysis of qualitative data will be conducted using Microsoft Word, and or alternative software such as NVivo (V.12). RC will analyse codes using NPT[44] and the BCTT[37] to focus on the social processes shaping the implementation process from making sense of the intervention to engaging in it (individually and collectively)

and embedding this (normalising) in everyday practice (or people's everyday lives) and to identify the facilitators and barriers present throughout the implementation process. Themes will be developed, from the codes, to reflect the lived experience and views of patient participants, health coaches and HCPs regarding the digital HC and mobile health platform intervention. MS will code 25% of interviews and focus groups separately to develop, discuss and agree on themes with RC. Finally, members of the core research team (EAS, MB, CC and JM) will be consulted to develop, discuss and agree on themes via an iterative process.

### Patient and public involvement
PPI and engagement have been and will continue to be part of all stages of the research from developing the proposal to providing feedback on preliminary findings, to dissemination and impact, and input for tasks such as finalising interview/focus group topic guides. For the development of this programme of research, we consulted with seven older adults and family carers who helped inform the proposal. They thought that this was an important topic and liked the HC approach to support digital engagement. We have convened a PPI advisory group (6–10 participants, aged 60+ years, meeting 3–4 times throughout the project). A PPI representative as a coresearcher leads the PPI group and attends the project advisory group. The group represents the views of people from different socioeconomic backgrounds including those who do and do not have access to various online and digital technologies.

PPI meetings will be face to face, and advisors will be offered reimbursement aligning with National Institution for Health and Care Research payment guidance. Advisors will have the opportunity to contribute via email and phone between meetings. Advisors will be sent materials prior to meetings and able to discuss matters arising from meetings afterwards. Advisors will be provided with meeting summaries and feedback on how their involvement informed the research project. A PPI plan will be developed and shaped by PPI and results/activities will be shared with the project advisory group. Monitoring and reporting will use the GRIPP checklist.[60]

### ETHICS AND DISSEMINATION
The DIALOR project is sponsored by Bournemouth University. Ethical approval has been granted by the Southeast Scotland Research Ethics Committee 02 (reference: 22/SS/0064). Data collected during the study will be pseudonymised by a member of the research team (RC) to ensure the removal of identifying information from research data prior to data analysis. Data will be stored on Bournemouth University servers, and identification data will be stored in encrypted files in a separate, secure location. Data transfer will be undertaken through secure, encrypted emails or Bournemouth University's secure file transfer system (BU Transfer). We will publish results in peer-reviewed open access journals. A lay summary of the final study report will be accessible on the project website: (https://www.bournemouth.ac.uk/

research/projects/dialor-digital-coaching-frailty). We will also seek to present our findings at conferences, workshops for patients, write lay articles for dissemination to the public and share the findings through our networks across health and social care.

## Status

The trial commenced in September 2023 and is expected to be completed by September 2024.

### Author affiliations

¹Faculty of Health and Social Sciences, Bournemouth University, Bournemouth, UK
²NHS Dorset, Poole, UK
³Wessex Academic Health Science Network, Chilworth, UK
⁴Faculty of Health Sciences, University of Southampton, Southampton, UK
⁵Older Person Medicine, University Hospitals Dorset NHS Foundation Trust, Bournemouth, UK
⁶Faculty of Environmental and Life Sciences, School of Health Sciences, University of Southampton, Southampton, UK
⁷The Adam Practice, Poole, UK
⁸School of Psychology, University of Southampton, Southampton, UK

**Acknowledgements** The authors thank study participants and patient advisers for their input. The authors wish to thank GP director Simone Yule for helping to implement the study within local primary care services. The authors would like to thank colleagues from Dorset Help & Care charity and my mhealth for helping to implement the study intervention and imbed this seamlessly within primary care services.

**Contributors** The study concept and design were conceived by JM. MIS, RC, EAS, CC and MB assisted in refining the study questionnaires and study design. MIS, RC and MA-P will be responsible for data collection. Analyses will be conducted by MIS, RC, EAS, D-MW, CO'S, KB, CC, MB, JM and MA-P. MIS prepared the first draft of the manuscript. All authors critically revised the manuscript and approved the submitted version. The corresponding author attests that all listed authors meet authorship criteria and that no others meeting the criteria have been omitted.

**Funding** This research was support by NIHR Applied Research Collaborative Wessex, grant number (ARC 001).

**Competing interests** None declared.

**Patient and public involvement** Patients and/or the public were involved in the design, or conduct, or reporting, or dissemination plans of this research. Refer to the Methods section for further details.

**Patient consent for publication** Not applicable.

**Provenance and peer review** Not commissioned; externally peer reviewed.

**ORCID iDs**
Matthew I Sait http://orcid.org/0000-0003-0226-0351
Sarah Thomas http://orcid.org/0000-0002-9501-9091

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
