## [Reviewer comments · BMJ Open]

ARTICLE DETAILS

TITLE (PROVISIONAL)	DIALOR (DIgitAL cOaching for fRailty): Protocol for a single-arm mixed methods feasibility study of a digital health coaching intervention for older people with frailty in primary care.
AUTHORS	Sait, Matthew; Christie, Rachel; Cox, Chantel; Board, Michele; Thomas, Sarah; O'Sullivan, Cheryl; Davies, Cheryl; Walker, Dawn-Marie; Vassallo, Michael; Sadler, EA; Allen-Pick, Mark; Moore, Patrick; Bradbury, Katherine; Murphy, Jane

VERSION 1 – REVIEW

REVIEWER	Masella, Cristina Politecnico di Milano
REVIEW RETURNED	25-Nov-2023

GENERAL COMMENTS	Dear authors the study protocol is surely interesting. It is well described and makes it possible to understand how data will be collected and processed. The number of patients is not very high and it is therefore suggested to be very careful about drop-outs. Being a single-arm study, many confounding factors will have to be dealt with when analysing the results. First of all the functioning of the platform and the unavoidable fine-tuning work that will be done. In any case, the world of digital health needs studies such as these to better understand how the services offered to doctors and patients work.
--

REVIEWER	Kouroubali, Angelina Foundation of Research and Technology Hellas, Institute of Computer Science
REVIEW RETURNED	10-Jan-2024

GENERAL COMMENTS	The paper tackles a very important problem, frailty management in older adults. The protocol aims to assess the acceptance and feasibility of an intervention. In line 46 page 6, the intervention development is described. How long will the development take? Where is the data coming from for the behavioural analysis? Further, in order to obtain meaningful measures, it is important to identify a clear timeline for the frequency of contact with the health coaches, and the types of goals that are set, for example, social goals, exercise goals, etc. It is suggested to take a baseline assessment for all measures before the intervention (including a comprehensive geriatric assessment) and then take measurements in 6 months and 12 months. There are also standardized tests for usability such as SUS, and others for technology acceptance that could be used to measure
---

	acceptability of the intervention. It is also important to elaborate on the study limitations such as bias about recruited older adults. etc.
--	---

VERSION 1 – AUTHOR RESPONSE

Reviewer 1		
	Comment	Response
1	the study protocol is surely interesting. It is well described and makes it possible to understand how data will be collected and processed. The number of patients is not very high and it is therefore suggested to be very careful about drop-outs.	Thank you. This is designed to be a feasibility study and aims to recruit numbers of participants in align with previous studies of sample sizes – refer to ref 42.
2	Being a single-arm study, many confounding factors will have to be dealt with when analysing the results. First of all the functioning of the platform and the unavoidable fine-tuning work that will be done. In any case, the world of digital health needs studies such as these to better understand how the services offered to doctors and patients work.	Thank you for your helpful comments. We are hopeful that this new digital health coaching model will provide new knowledge and approach to help support the management of frailty in its early stages, alongside another long-term health condition, and better understanding of the barriers and facilitators to support implementation.
Reviewer 2		
	Comment	Response
1	The paper tackles a very important problem, frailty management in older adults. The protocol aims to assess the acceptance and feasibility of an intervention. In line 46 page 6, the intervention development is described. How long will the development take? Where is the data coming from for the behavioural analysis?	Thank you for your helpful comments about the intervention and have indicated that the development will take 9 months. More details for the behavioural analysis have been added on page 6.
2	Further, in order to obtain meaningful measures, it is important to identify a clear timeline for the frequency of contact with the health coaches, and the types of goals that are set, for example, social goals, exercise goals, etc. It is suggested to take a baseline assessment for all measures before the intervention (including a comprehensive geriatric assessment) and then take measurements in 6 months and 12 months. There are also standardized tests for usability such as SUS, and others for technology acceptance that could be used to measure acceptability of the intervention. It is also important to elaborate on the study limitations such as bias about recruited older adults. etc.	Thank you. Participants will receive approximately 12 weeks of health coaching with 6 sessions fortnightly, these sessions will each be 45 mins – 1 hour in duration. The participants will also have access to a mobile health platform for 6 months which will include access to the particular application in relation to their long-term condition. In the text we have indicated the number of times health coaches will engage with

	older people with frailty and a long-term condition, see page 4. We have also amended the study limitations in the summary and referred to limitations associated with the representation of the recruited older adults and generalisability.
--	---

VERSION 2 – REVIEW

REVIEWER	Masella, Cristina Politecnico di Milano
REVIEW RETURNED	03-Mar-2024

GENERAL COMMENTS	Dear Authors, I find your research protocol very interesting and useful. My only concern is about confounding factors that are related to disease progression and/or social context. It is not so clear from the protocol how you will handle them. Furthermore, it is not specified how you will choose from the 50 patients in the study the 20 who will be interviewed and why 20 and not the entire study population
---

VERSION 2 – AUTHOR RESPONSE

Manuscript status update on bmjopen-2023-080480

Title: DIALOR (DigtAL cOaching for fRailty): Protocol for a single-arm mixed methods feasibility study of a digital health coaching intervention for older people with frailty in primary care.

We would like to thank the editor and reviewers for giving their time to review our manuscript and for their positive and insightful comments.

Reviewer 1		
	Comment	Response
1	My only concern is about confounding factors that are related to disease progression and/or social context. It is not so clear from the protocol how you will handle them.	Thank you for this comment. At the current stage, this is designed to be an acceptability and feasibility study. Considering this, the study currently aims to answer whether this intervention can be done, should we proceed with it, and if so, how. Further, owing to

		the multifaceted and complex study design; we are evaluating the intervention in accordance with the medical research councils (MRC) complex interventions framework. Inevitably, this hard work will underpin a larger piece of research. We feel that at future stages, when evaluating intervention effectiveness, we will begin to be able to control for confounding factors, in our data analysis. However, owing to the nature of the acceptability and feasibility design, we do not feel that controlling for confounding factors is possible, at this stage. We will delightedly carry your feedback forwards and apply it to our future work.
2	Furthermore, it is not specified how you will choose from the 50 patients in the study the 20 who will be interviewed and why 20 and not the entire study population	Thank you for the helpful comment. We will undertake purposive sampling to select older participants (following ethics protocol for consent), up to 20 semi-structured interviews or until data saturation is reached, in accordance with robust methods in qualitative research (Francis et al 2010), refer to ref 38, and similar studies that have undertaken interviews with older people with frailty.